# Improved Production of Mashua (*Tropaeolum tuberosum*) Microtubers MAC-3 Morphotype in Liquid Medium Using Temporary Immersion System (TIS-RITA®)

**Gilmar Peña-Rojas** [1] **, Roxana Carhuaz-Condori** [1] **, Vidalina Andía-Ayme** [2]**, Victor A. Leon** [3]
**and Oscar Herrera-Calderon** [4],*

1   Laboratory of Cellular and Molecular Biology, Biological Sciences Faculty, Universidad Nacional de San Cristóbal de Huamanga, Portal Independencia 57, Ayacucho 05003, Peru; gilmar.pena@unsch.edu.pe (G.P.-R.); roxana.carhuaz@unsch.edu.pe (R.C.-C.)
2   Laboratory of Food Microbiology, Biological Sciences Faculty, Universidad Nacional de San Cristóbal de Huamanga, Portal Independencia 57, Ayacucho 05003, Peru; vidalina.andia@unsch.edu.pe
3   Department of Biology, New York University, New York, NY 10003, USA; victorleon@nyu.edu
4   Department of Pharmacology, Bromatology and Toxicology, Pharmacy and Biochemistry Faculty, Universidad Nacional Mayor de San Marcos, Lima 15001, Peru
*   Correspondence: oherreraca@unmsm.edu.pe; Tel.: +51-956-550-510

**Abstract:** Essential molecules are embedded within the millenary crop *Tropaeolum tuberosum* (mashua); these compounds are critical for the Andean people's traditional diet and extensively utilized by the pharmaceutical industry in Peru. In the Andean region, conventional cropping techniques generate microtubers susceptible to a viral infection, which substantially endangers mashua's production. Therefore, we developed an innovative in vitro technique condition for enhancing the agriculture process for micro tubers production. The temporary immersion system (TIS) permits the production of high-quality microtubers in a reduced space, a lower amount of time, and in large quantities compared with tubers grown under traditional conditions. To obtain *T. tuberosum*'s microtubers via TIS, we propagated seedlings, utilizing TIS-RITA® vessels. A set of immersion frequency times were evaluated. Interestingly, results showed that immersion at 2 min every 3 h was more beneficial compared with 2 min every 5 h based on microtubers produced after 10 weeks from the treatments, revealing an efficient frequency setting which outputted improved microtubers quality and production.

**Keywords:** microtubers; temporary immersion system; *Tropaeolum tuberosum*; mashua

## 1. Introduction

The mashua is a millenary crop that contains substantial nutritional and medical properties [1–3]. This crop originated from the Andean region [4], and is considered the fourth most crucial Andean root among other tubers such as potatoes, oca, and olluco [5]. The mashua is a propagation crop cultivated over the latest centuries across the Andean mountains in Peru, Bolivia, Ecuador, Venezuela, and Colombia [5,6]. This formidable tuber has managed to grow under nutrient-deprived soil conditions and at high altitudes without fertilizers or pesticides, outstanding for its resistance against harsh conditions in contrast to other contemporary crops [7,8].

Traditionally, the Andean mashua is propagated for production purposes as other tubers within the Andean crop fields [9]. Additionally, when the mashua is cultivated under field conditions, it necessitates between 6 and 8 months to properly attain its vegetative cycle stages, and in several cases, the tubers become virally infected, dramatically impacting the crop's production [6]. Therefore, rural and local mashua production within the Andean region does not ensure suitable seed quality and necessitates alternative cropping techniques to improve biological features such as growth and vigor. In vitro techniques

are utilized to improve crop production and decrease time constraints; therefore, sculpting innovative in vitro techniques for tuber cropping is needed.

One in vitro technique to harness the growth of seedlings is the modulation of the frequency and duration of immersion times [10] during the tuber early development. The TIS (temporary immersion system) strategy enables the rapid and efficient propagation of several plants with keen agricultural interest. The TIS enhances the growing speed and ensures the optimal quality of the plant tissue generated in vitro [11]. The TIS permits the production of high-quality pathogen-free seedlings and microtubers in vitro at any time throughout the year [12]. Moreover, it reduces large-scale crop production costs [13], automatizes the cropping process, and permits proper propagation by utilizing liquid media to ensure seedlings vigor [11].

The TIS enhances the growing speed and ensures the optimal quality of the plant tissue generated in vitro [11], enabling rapid and efficient propagation of several plants with strong agricultural interest. The TIS technique initiates by inducing air pressure flow through an air compressor; this de novo pressure elevates the liquid media permitting contact with the explants localized inside the chamber intermittently. As the air injection subsides within the system, and the media descends by gravity, the atmosphere remodels within the system, facilitating a robust growth and substantial improvement for the seedlings' development [14]. Some critical factors defining the TIS technique are the following: the optimization of the total volume inside the vessels, the supplement in the media, the vitrification, the ethylene accumulation, and the carbon dioxide. Seedlings development and growth can be harnessed by modifying the frequency and duration of the immersion time [10].

Likewise, TIS-RITA® includes a structured container divided into two vessels: a superior vessel hosting the plants and an inferior vessel containing the media. The applied overpressure to the inferior vessel propels the media towards the superior compartment generating bubbles grazing the plant tissues. At this stage, seedlings temporarily submerge as overpressure is delivered. During the immersion period, the media falls by gravity. One result is the altered atmosphere inside the container [12]. The other critical parameter is the immersion time, involved in efficient sprout micropropagation, microtuberization, and somatic embryogenesis [12]. The TIS is regularly utilized for increasing in vitro propagation coefficients compared with field conditions. For example, these methods have been in used in other crop species such as bananas [15], anthurium [16], sugar cane [17], and potato microtubers [18].

In this study, we concentrated on the production of mashua MAC-3 morphotype via a novel in vitro procedure. MAC-3 morphotype (purple mashua) is known for its high antioxidant activity, total phenolic, tannins, total flavonoids, and total anthocyanins [19]. Recently, we had faithfully propagated *Solanum tuberosum*, *Oxalis tuberosa*, and *Ullucus tuberosum* in vitro utilizing TIS-RITA [20]. In a previous report, we found that a specific frequency condition in TIS-RITA substantially enhances the mashua's microtuber propagation [21]. Therefore, we explored vital settings to generate high-quality seeds, utilizing a set of immersion frequencies to obtain improved seeds of the *T. tuberosum* MAC-3 morphotype that would positively impact the crop production for the Andean community in South America.

## 2. Materials and Methods

### 2.1. Micropropagation Study

Mashua seedlings (*T. tuberosum* Ruiz & Pav.) derived from a MAC-3 morphotype from the germplasm bank of the Cellular and Molecular Biology Laboratory (UNSCH, Ayacucho, Peru) were propagated in vitro (Figure 1). Seedlings were maintained using Murashige and Skoog 1962 (MS) solid medium. After 30 days of culture, the seedlings grown in solid medium were transferred to flasks containing 100 mL of MS liquid medium supplemented with 3% sucrose at a pH of 5.6 to obtain seedling vigor; the flasks were kept under constant

agitation on an orbital shaker. Culture conditions were 19 ± 2 °C; 16 h of light and 8 h of darkness with a relative humidity between 60% and 70% during the multiplication phase.

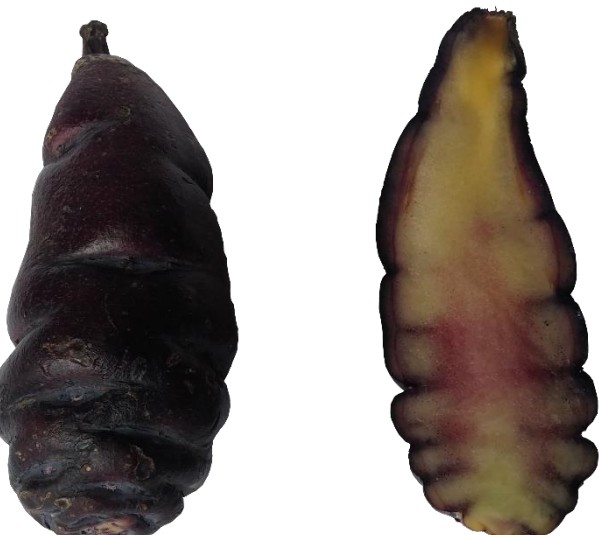

**Figure 1.** *T. tuberosum* Ruiz and Pav. "mashua" MAC-3 morphotype used in the TIS to obtain microtubers.

### 2.2. Production of Microtubers via TIS-RITA

TIS-RITA vessels were used according to Etienne et al. [12]. Murashige and Skoog (MS) liquid medium were prepared inside the RITA vessels, supplemented with 2 ppm BAP and 8% sucrose, at a pH 5.6; the vessels were sterilized at 121 °C for 15 min. This study evaluated one immersion time point and two frequencies: 2 min every 3 h and 2 min every 5 h. Next, TIS-RITA vessels were incubated under the constant temperature of 19 °C ± 2 °C, for a total of 10 weeks in total darkness. Produced microtubers were harvested off the culture vessels, and samples were rinsed off with tap water to remove any excess media. These microtubers were placed on trays covered with filter paper to remove humidity. Finally, the microtubers' fresh weights (g) were evaluated using an analytical scale and a vernier ruler to measure the size (cm).

### 2.3. Data Analysis

The output data were statistically analyzed by using a random design format with double replicates. The variation analysis was performed to compare the size and weight of microtubers in a non-parametric U-Mann–Whitney test.

### 3. Results and Discussion

Using the RITA® temporary immersion system, it was possible to obtain mashua microtubers from MAC-3 morphotype using the two immersion frequencies. However, there was a significant difference between the two immersion frequencies in obtaining the size of the microtubers; a size of 1.09 cm was achieved at an immersion frequency of every three hours for two minutes, compared with 0.86 cm at a frequency of every five hours for two minutes respectively (Figure 2A), these findings being statistically significant ($p = 0.0017$; U-Mann Whitney test). According to Akita et al. [22], potato microtubers (*Solanum tuberosum* L.) were obtained using a laboratory-scale fermenter with a weight of more than 0.2 g. Montoya et al. [23] achieved the greatest number and size of *Solanum tuberosum* shoots using TIS with an immersion frequency of three hours. Escalona et al. [24] in the cultivation of *Ananas commosus* achieved a higher multiplication rate using an immersion time of two minutes and a frequency of three hours. Likewise, Cabrera et al. [25] obtained a greater number and size of *Dioscorea alata* microtubers with significant differences in relation to other immersion times using an immersion time of 15 min and after 18 weeks of culture. Etienne et al. [12]

stated that *Solanum tuberosum* microtubers and *Coffea arabica* somatic embryos produced in temporary immersion bioreactors developed satisfactorily after planting.

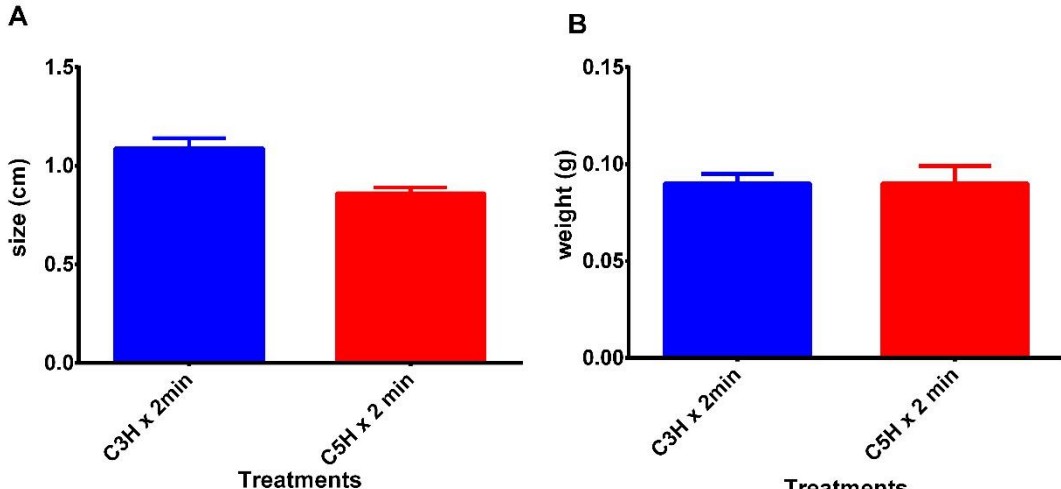

**Figure 2.** (**A**) Mean comparison of the microtuber size and (**B**) weight from *T. tuberosum* "mashua" obtained in two TIS treatments: 2 min every 3 h (C3H × 2 min); 2 min every 5 h (C5H × 2 min).

On the other hand, using a TIS, 52 mashua microtubers from a MAC-3 morphotype were obtained at a frequency of every three hours of immersion for two minutes compared with a frequency of five hours for two minutes in which 39 microtubers were obtained. Moreover, Igarza et al. [26] acquired an average of between five and seven potato microtubers of the "Andinita" variety using a TIS. Montoya et al. [23] achieved greater efficiency in the in vitro tuberization of *Solanum tuberosum* variety Diacol Capiro when used in temporary immersion bioreactors and in MS medium supplemented with 1 ppm of 6-Benzylaminopurine (BAP) and 8% sucrose; in addition, the microtubers obtained in a TIS allowed the formation of tubers under field conditions. Gopal et al. [27]) concluded that microtubers produced in media without abscisic acid (ABA) during and containing high concentrations of sucrose and BAP can be stored for 12 months.

Regarding the fresh weight of the mashua, an average of 0.09 g was achieved in both immersion frequency treatments; therefore, there was no statistically significant difference in the results obtained between both treatments (Figure 2B). Igarza et al. [25] achieved an average fresh weight that did not exceed 3.5 g using an immersion system to obtain potato microtubers cv. "Andinita".

The immersion system allows obtaining microtubers in two and a half months (Figure 3), significantly reducing the production time compared with the production of mashua tubers in the field, which generally requires between six to nine months. In addition, under this production system it is possible to obtain high-quality seeds, allowing ex situ conservation in a germplasm bank, and fundamentally for use in seed management and improvement programs.

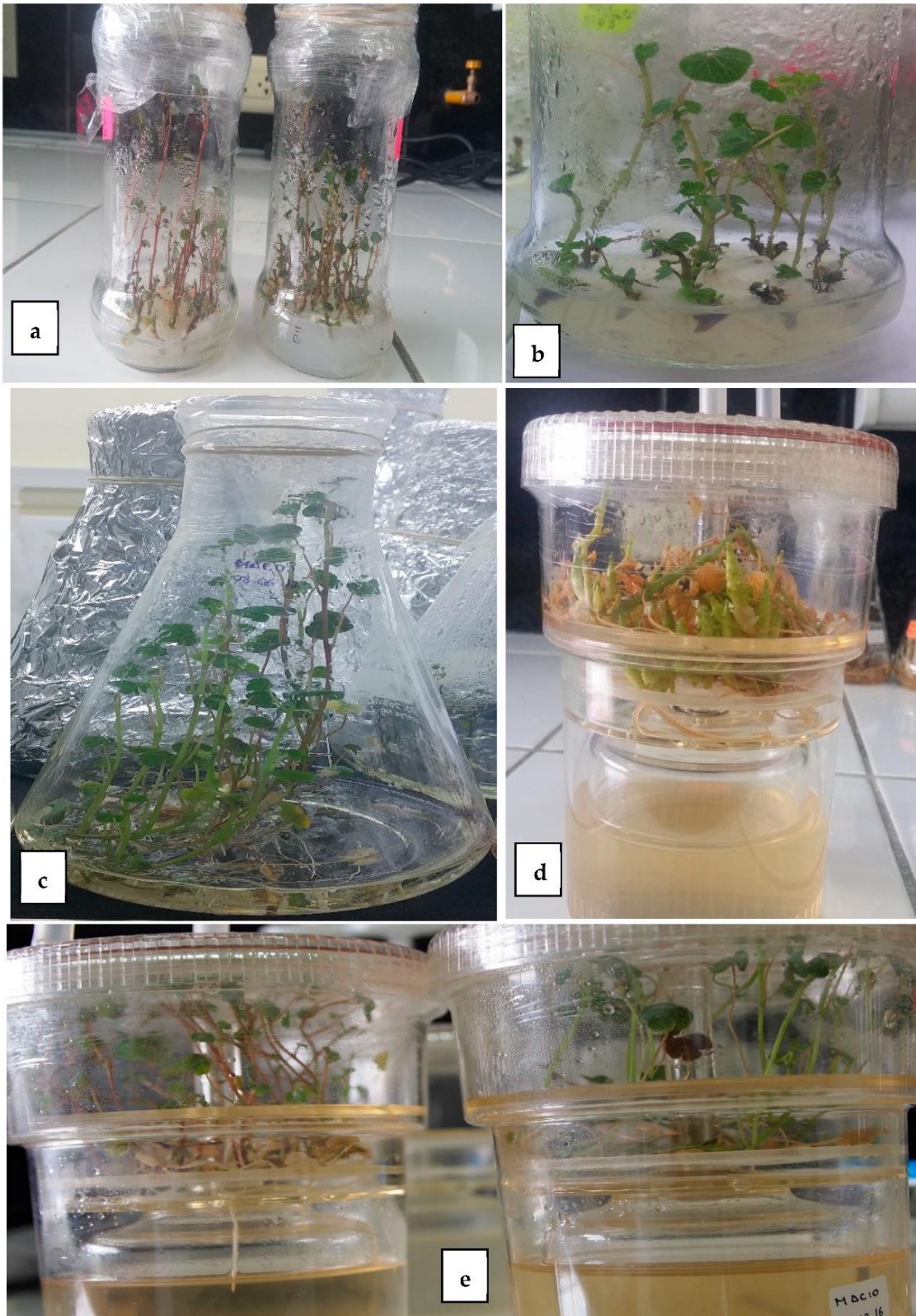

**Figure 3.** (**a**) Conservation in germplasm bank. (**b**) Micropropagation in solid medium. (**c**) Propagation in liquid medium. (**d**) Obtaining microtubers in a temporary immersion from *T. tuberosum* "mashua" MAC-3 after a 10-week culture. (**e**) The frequency: (Left): three hours for two minutes, (right): five hours for two minutes.

## 4. Conclusions

It was possible to obtain MAC-3 microtubers in the TIS RITA® using the Murashige and Skoog medium supplemented with 8% sucrose, 2 ppm BAP, and with an immersion frequency of every 3 h for 2 min. The TIS RITA® is an efficient alternative for the production of high-quality seeds. Furthermore, it would lead to obtaining virus-free microtubers as well as reducing the harvest time compared with traditional production techniques.

**Author Contributions:** Conceptualization, G.P.-R. and R.C.-C.; methodology, G.P.-R.; formal analysis, V.A.-A.; investigation, G.P.-R., R.C.-C., V.A.-A. and O.H.-C.; writing—original draft preparation, V.A.L.; writing—review and editing, V.A.L.; visualization, O.H.-C.; project administration, G.P.-R.; funding acquisition, G.P.-R. All authors have read and agreed to the published version of the manuscript.

**Funding:** This research was funded by the CONCYTEC, Ministerio de Educacion, Perú. (MINEDU-CONCYTEC) project 199-2015-FONDECYT—UNSCH.

**Institutional Review Board Statement:** Not applicable.

**Data Availability Statement:** Not applicable.

**Acknowledgments:** Authors would like to thank FONDECYT and the Department of Biology of the New York University.

**Conflicts of Interest:** The authors declare no conflict of interest.

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
