# Peer review of "Improved Production of Mashua (Tropaeolum tuberosum) Microtubers MAC-3 Morphotype in Liquid Medium Using Temporary Immersion System (TIS-RITA®)"

_agriculture, doi:10.3390/agriculture12070943_

Round 1

Reviewer 1 Report

The  TIS (temporary immersion system) for mashua propagation has been previously developed and described e.g. Peña-Rojas et al., 2020 Tropical and Subtropical Agroecosystems 23 (2020): #83, so what is the novelty of this report?

Author Response

Dear Reviewer 1,

The TIS (temporary immersion system) for mashua propagation has been previously developed and described e.g., Peña-Rojas et al., 2020 Tropical and Subtropical Agroecosystems 23 (2020): #83, so what is the novelty of this report?

R1. In the work: “USE OF RITA® TEMPORARY IMMERSION SYSTEM TO OBTAIN MICROTUBERS OF VARIOUS MORPHOTIPOS OF MASHUA (Tropaeolum tuberosum Ruiz & Pavón), only an immersion frequency of 2 minutes every 3 hours was increased because 27 mahua morphotypes were worked with. However, for the black mashua, which is one of the most important crops for containing higher levels of bioactive compounds, two immersion frequencies were tested: 2 minutes every 3 hours and 2 minutes every 5 to confirm if it improved in obtaining microtubers. Confirming that the optimal immersion time is every 3 hours for 2 minutes for 10 weeks to obtain a greater number of microtubers.

To increase the time of two frequencies, the following was taken as a reference:

Igarza et al., (2011) tested three immersion frequencies (every 2h, 3h and 4h) to obtain potato microtubers cv. ‘Andinita’ in Temporary Immersion Systems, being the best at 4 hours of immersion.

Tapia et al., (2017) used an immersion time and frequency of 4 minutes every 3 hours, to obtain microtubers Obtaining microtubers and minitubers as prebasic seed in three Peruvian potato cultivars.

Montoya et al., (2008) evaluated a single frequency of 3 hours and 3 minutes of immersion for in vitro tuberization of potatoes.

Iles (2015) evaluated 3 minutes every 12 hours and 3 minutes every 4 hours for the microtuberization of INIAP potato cultivars.

References:

1. Igarza, J. I., Agramonte, D., de Feria, M., Jaime, J., Pérez, M., & San Román, M. (2011). Obtención de microtubérculos de papa cv.‘Andinita’en Sistemas de Inmersión Temporal. Biotecnología Vegetal, 11(1).

2. Tapia M., Lorenzo J., Mosqueda O. y Escalona M. (2017). Obtention of microtubers and minitubersas pre basic seed in tree Peruvian potatocultivars. Laboratorio de Biotecnología, Universidad Naciona la Agraria. Centro de Bioplantas, Universidad de Ciego.

3. Montoya N., Castro D., Díaz J. y Ríos D. (2008) Tuberización in vitro de papa (Solanum tuberosum L), variedad Diacol Capiro, en biorreactores de inmersión temporal y evaluación de su comportamiento en campo. Unidad de Biotecnología Vegetal. Universidad Católica de Oriente. Colombia. CIENCIA 16(3), 288 – 295.

4. Iles D., Piedra M., Orbe K., y Morillo E. (2015). Microtuberización de los cultivares de papa INIAP-Victoria y Supechola bajo sistemas de inmersión temporal. Memorias del VI Congreso Ecuatoriano de la Papa (pp. 122-124). Ibarra, Ecuador: INIAP/CIP.

Reviewer 2 Report

The manuscript with the title “Improved production of mashua (Tropaeolum tuberosum) microtubers in liquid medium using temporary immersion system RITA®” is a short communication that presents comparatively the results of two variants (represented by different immersion time frequencies) for in vitro production of microtubers of mashua. Mashua is an interesting niche edible crop from Peru.
The paper is interesting but would benefit from editing for clarity in several places.

Abstract
Line 24 says “mashua microtubers were generated every 3 hours 24 for 2 minutes of frequency”, but by looking at Material and Method I would suggest the re-writing of the phrase such as:
“Results showed that immersion at 2 minutes every 3 hours was more beneficial compared to 2 minutes every 5 hours based on microtubers produced after 10 weeks from the treatment”.

Introduction – the aim
Lines 83-85, it is stated that generation of seed was the aim, but in fact microtubers were produced (yes, both are planting material but in this case were microtubers from what I undertsnad?).

Author Response

Dear Reviewer 2

The manuscript with the title “Improved production of mashua (Tropaeolum tuberosum) microtubers in liquid medium using temporary immersion system RITA®” is a short communication that presents comparatively the results of two variants (represented by different immersion time frequencies) for in vitro production of microtubers of mashua. Mashua is an interesting niche edible crop from Peru.

The paper is interesting but would benefit from editing for clarity in several places.

Thank you for your valuable comments with our article.

Abstract

Line 24 says “mashua microtubers were generated every 3 hours 24 for 2 minutes of frequency”, but by looking at Material and Method I would suggest the re-writing of the phrase such as:
“Results showed that immersion at 2 minutes every 3 hours was more beneficial compared to 2 minutes every 5 hours based on microtubers produced after 10 weeks from the treatment”.

R1. Thank you for your suggestions, it was added in the abstract.

Introduction – the aim. Lines 83-85, it is stated that generation of seed was the aim, but in fact microtubers were produced (yes, both are planting material but in this case were microtubers from what I undertsnad?).

R2: Thank you for your observation, generally in the agriculture, they use mashua tubers as seed for planting or cultivation. However, these seeds contain different viruses that decrease productivity. Therefore, in the laboratory under in vitro culture conditions, work was carried out on virus cleaning by meristem culture to obtain virus-free microtubers that would be marketed as high-quality seeds in the market.

Reviewer 3 Report

The article entitled "Improved production of mashua (Tropaeolum tuberosum) microtubers in liquid medium using temporary immersion system RITA®" is well written and of great economic value, this technique which was applied successfully in the past to other crops and is now applied on mashua (Tropaeolum tuberosum). The authors should highlight if the cost of this technique could be compensated by a higher yield in the crops.

in the introduction in the first sentence the authors said:The mashua is a millenary crop that contains substantial nutritional and medical properties" they should provide some references, I would like to suggest the following:

Ticona, L.A., Sánchez, Á.R., Estrada, C.T. and Palomino, O.M., 2021. Identification of TRPV1 Ion Channels Agonists of Tropaeolum tuberosum in Human Skin Keratinocytes. Planta Medica87(05), pp.383-394.

Ticona, L.A., Sebastián, J.A., Serban, A.M. and Sánchez, Á.R., 2020. Alkaloids isolated from Tropaeolum tuberosum with cytotoxic activity and apoptotic capacity in tumour cell lines. Phytochemistry177, p.112435.

Ticona, L.N.A., Pérez, V.T. and Benito, P.B., 2020. Local/traditional uses, secondary metabolites and biological activities of Mashua (Tropaeolum tuberosum Ruíz & Pavón). Journal of ethnopharmacology247, p.112152.

Author Response

Dear Reviewer 3.

The article entitled "Improved production of mashua (Tropaeolum tuberosum) microtubers in liquid medium using temporary immersion system RITA®" is well written and of great economic value, this technique which was applied successfully in the past to other crops and is now applied on mashua (Tropaeolum tuberosum). The authors should highlight if the cost of this technique could be compensated by a higher yield in the crops.

In the introduction in the first sentence the authors said: The mashua is a millenary crop that contains substantial nutritional and medical properties" they should provide some references; I would like to suggest the following:

Ticona, L.A., Sánchez, Á.R., Estrada, C.T. and Palomino, O.M., 2021. Identification of TRPV1 Ion Channels Agonists of Tropaeolum tuberosum in Human Skin Keratinocytes. Planta Medica87(05), pp.383-394.

Ticona, L.A., Sebastián, J.A., Serban, A.M. and Sánchez, Á.R., 2020. Alkaloids isolated from Tropaeolum tuberosum with cytotoxic activity and apoptotic capacity in tumour cell lines. Phytochemistry177, p.112435.

Ticona, L.N.A., Pérez, V.T. and Benito, P.B., 2020. Local/traditional uses, secondary metabolites and biological activities of Mashua (Tropaeolum tuberosum Ruíz & Pavón). Journal of ethnopharmacology247, p.112152.

R1. Thank you so much for your comments, we added those references in our article.

Round 2

Reviewer 1 Report

The manuscript requires improvements:

Line 100: seedlings really grown under medium? Or were cultivated on solid MS medium;

Lines 104-122? “The total incubation period had a 30-day total duration. Seedlings grew in the solid  medium were transferred to 100mL Erlenmeyer flasks with supplemental media. The media contents included MS liquid media supplemented with 3% sucrose pH 5.6. To augment seedlings’ vigor, the flask’s contents were placed inside an orbital shaker.” – this is confusing, what mean “incubation” – duration of culture necessary to induce microtubes? The same in lines 133-134;

Seedlings on solid medium grew on it;  and the cultures in liquid media have to be carried out on shaker which is substantial to air transfer into medium;

Line 127: description is…… described: please correct;

Line 130: 15 pounds? – this is not SI unit;

“We evaluated two immersion time points and frequency: 2 minutes every 3 hours and 2 minutes every 5 hours.” In fact this is one immersion time and two frequencies;

Section: Results and discussion: this part should present results of present study first, afterwards discuss the results of previous investigations. This section requires substantial improvements and should be re-write.

Lines 143-144: “The TIS displayed remarkable micropropagation development, strengthened microtuberization, normal somatic embryogenesis, which suggested that TIS enhances the molecular behavior of cropping species in vitro [12].” – it is not clear what about and what for this is put at the beginning of this section - please re-write;

“Induced mashua microtubers produced morphotype MAC3 seedlings by applying two immersion frequencies to improve growth and physical features.” – nothing is said about regeneration of seedlings from induced microtubes before i.e. in Materials and Methods.

“The first treatment consisted of 2 minutes every 3 hours frequency and outputted an average of 1.09 cm.” – what about is this? It is incomprehensible for the readers;

Line 169: micortubes microtues – please correct; and what for is this: “U de Mann-Whitney test; U=749.000, α=0.05, Sig. =0.017).” – it should be clearly stated if the differences were or not statistically significant;

Figure 2: The captions A and B have to be correct: comparison between effect of TIS treatments on microtube size e.g.; and in similar manner B;

What is ”nutritional supplement”?

Just Montoya et al., without In;

The rest of this section has to be corrected.

Conclusions should be based on only on results of present study.

Author Response

Reviewer 1:

The manuscript requires improvements:

  1. Line 100: seedlings really grown under medium? Or were cultivated on solid MS medium;

R1: The MAC 3 tuber of T. tuberosum Ruiz & Pav was initially grown in solid MS medium, then the seedlings were transferred to liquid MS medium using Erlenmeyer flasks on an orbital shaker to increase seedling vigor. Finally, these seedlings grown in liquid medium and then they were transferred to TIS RITA containers to obtain microtubers.

  1. Lines 104-122? “The total incubation period had a 30-day total duration. Seedlings grew in the solid medium were transferred to 100 mL Erlenmeyer flasks with supplemental media. The media contents included MS liquid media supplemented with 3% sucrose pH 5.6. To augment seedlings’ vigor, the flask’s contents were placed inside an orbital shaker.” – this is confusing, what mean “incubation” – duration of culture necessary to induce microtubes? The same in lines 133-134;

R2. It was corrected and the protocol was rewritten.

  1. Line 127: description is …… described: please correct;

R3. This sentence was corrected.

  1. Line 130: 15 pounds? – this is not SI unit;

R4. Thank you for your observation, it was excluded in the methodology. We do not use pressure in the methodology.

  1. “We evaluated two immersion time points and frequency: 2 minutes every 3 hours and 2 minutes every 5 hours.” In fact, this is one immersion time and two frequencies;

R5: Thank you, it was modified according to your suggestions.

  1. Section: Results and discussion: this part should present results of present study first, afterwards discuss the results of previous investigations. This section requires substantial improvements and should be re-write.

R6: Thank you, it was modifed.

  1. Lines 143-144: “The TIS displayed remarkable micropropagation development, strengthened microtuberization, normal somatic embryogenesis, which suggested that TIS enhances the molecular behavior of cropping species in vitro [12].” – it is not clear what about and what for this is put at the beginning of this section - please re-write;

R7: It was modified.

  1. “Induced mashua microtubers produced morphotype MAC3 seedlings by applying two immersion frequencies to improve growth and physical features.” – nothing is said about regeneration of seedlings from induced microtubes before i.e. in Materials and Methods.

R8: It was corrected.

  1. “The first treatment consisted of 2 minutes every 3 hours frequency and outputted an average of 1.09 cm.” – what about is this? It is incomprehensible for the readers;

R9: It was corrected.

  1. Line 169: micortubes microtues – please correct; and what for is this: “U de Mann-Whitney test; U=749.000, α=0.05, Sig. =0.017).” – it should be clearly stated if the differences were or not statistically significant;

R10: It was corrected.

  1. Figure 2: The captions A and B have to be correct: comparison between effect of TIS treatments on microtube size e.g.; and in similar manner B;

R11: Figures were modified.

  1. What is ”nutritional supplement”?

R12: It was amended.

13. Just Montoya et al., without In;

R13: It was amended.

14. The rest of this section has to be corrected.

R14: It was corrected the methodology and discussion.

15. Conclusions should be based on only on results of present study.

R15: It was corrected and rephrased.